# Improvement of the Seminal Characteristics in Rams Using Agri-Food By-Products Rich in Phytomelatonin

**DOI:** 10.3390/ani13050905

**Published:** 2023-03-02

**Authors:** Victoria Peña-Delgado, Melissa Carvajal-Serna, Manuel Fondevila, María A. Martín-Cabrejas, Yolanda Aguilera, Gerardo Álvarez-Rivera, José A. Abecia, Adriana Casao, Rosaura Pérez-Pe

**Affiliations:** 1Grupo BIOFITER-Instituto Universitario de Investigación en Ciencias Ambientales de Aragón (IUCA), Facultad de Veterinaria, Universidad de Zaragoza, 50013 Zaragoza, Spain; 2Departamento de Producción Animal y Ciencia de los Alimentos, Instituto Agroalimentario de Aragón (IA2), Universidad de Zaragoza-CITA, 50013 Zaragoza, Spain; 3Departamento de de Química Agrícola y Bromatología, Facultad de Ciencias, Universidad Autónoma de Madrid, 28049 Madrid, Spain; 4Instituto de Investigación en Ciencias de la Alimentación, Spanish National Research Council (CSIC)-Universidad Autónoma de Madrid (UAM), 28049 Madrid, Spain

**Keywords:** circular economy, diet, melatonin, phytomelatonin, ram, seminal quality

## Abstract

**Simple Summary:**

One of the limiting factors in sheep husbandry is reproductive seasonality, which is regulated by nocturnal melatonin secretion. Subcutaneous implants of this hormone have been used to modulate this seasonality. Nowadays, consumers are increasingly concerned about organic, hormone-free production. In order to adapt sheep production to these new demands, it would be of interest to replace synthetic melatonin with phytomelatonin, which is present in plants and can be included in sheep diet. In addition, if phytomelatonin comes from by-products from the food industry, a further step would be taken towards the objectives of the circular economy. Thus, the main objective of this work was to evaluate the effect of phytomelatonin-rich diets on ram sperm quality and seminal plasma composition. With this work, we found that a phytomelatonin-rich diet, including a mix of grape pulp, and pomegranate and tomato pomaces, can increase melatonin levels in seminal plasma, improve sperm viability and morphology, and protect sperm cells against oxidative damage.

**Abstract:**

The aim of this study was to evaluate the effect of a phytomelatonin-rich diet, including by-products from the food industry, on ram sperm quality and seminal plasma composition. Melatonin content in several by-products before and after in vitro ruminal and abomasal digestion was determined by HPLC-ESI-MS/MS. Finally, 20% of a mix of grape pulp with pomegranate and tomato pomaces was included in the rams’ diet, constituting the phytomelatonin-rich diet. Feeding the rams with this diet resulted in an increase in seminal plasma melatonin levels compared with the control group (commercial diet) in the third month of the study. In addition, percentages higher than those in the control group of morphologically normal viable spermatozoa with a low content of reactive oxygen species were observed from the second month onwards. However, the antioxidant effect does not seem to be exerted through the modulation of the antioxidant enzymes since the analysis of the activities of catalase, glutathione reductase and glutathione peroxidase in seminal plasma revealed no significant differences between the two experimental groups. In conclusion, this study reveals, for the first time, that a phytomelatonin-rich diet can improve seminal characteristics in rams.

## 1. Introduction

Livestock farming systems in the Mediterranean regions of the southern European Union countries, such as sheep husbandry, are important [1] given that they are linked with the use of semi-natural and natural areas, and involve well-adapted autochthonous breeds [2], which is the case for the Rasa Aragonesa breed [3]. However, these systems are currently threatened by economic, institutional, environmental and social factors [4]. Furthermore, one of the limiting factors is reproductive seasonality, which is regulated by nocturnal melatonin secretion [5] in the pineal gland. Melatonin is also synthesized in the male reproductive tract [6] and is present in seminal plasma [7], having direct effects on ram sperm functionality. Experiments conducted with melatonin added in vitro to ovine sperm samples have demonstrated that this hormone. modulates sperm capacitation, decreases oxidative stress and apoptosis markers [8,9,10].

Regarding the in vivo effects of melatonin on small ruminants, studies in the 1980s were performed by orally administering melatonin [11,12,13]. This hormone, absorbed onto food pellets or added in saline solution, resulted in sustained elevated blood levels of melatonin for at least 7 h in ewes [11]. Moreover, Arendt et al. reported that daily oral administration allowed advancing the reproductive season in ewes [12] although the efficacy of this treatment decreased if the administration was reduced to three times a week [13]. Most of these experiments were performed on ewes, but few studies examined the effects of the oral administration of melatonin on rams. Among these, the work performed on Suffolk rams by Kusakari and Ohara revealed that melatonin feeding could increase the reproductive activity of these rams during periods of seasonal regression [14].

Since the 1990s, with the development of melatonin subcutaneous implants, the use of these devices has displaced studies with oral melatonin, as they are more practical [15,16,17,18,19,20,21,22,23]. In this case, our group and others demonstrated that implants had beneficial effects on reproductive traits, sperm motility and fertilization parameters, the seminal plasma hormonal profile and the activity of some antioxidant enzymes in rams [15,16,17,18,19]. There is abundant research into the effects of melatonin implants on the semen of other ruminants [20,21,22,23].

However, in order to adapt sheep production to the new demands of consumers, who are increasingly concerned about organic, hormone-free production, it would be of interest to replace the synthetic melatonin present in implants with other natural sources of melatonin such as phytomelatonin, which is the melatonin present in plants and which can be administered with the diet.

Plants contain phytomelatonin in highly variable concentrations [24]. The highest levels have been found in seeds such as mustard (*Brassica nigra* and *Brassica hirta*, 129 and 189 ng/g of dry matter, respectively), goji (*Lycium barbarum,* 103 ng/g), fenugreek (*Trigonella foenum-graecum,* 43 ng/g), almond (*Prunus amygdalus,* 39 ng/g), sunflower (*Helianthus annuuss,* 29 ng/g), fennel (*Foeniculum vulgare,* 28 mg/g) and alfalfa *(Medicago sativum,* 16 ng/g) [25], and also in fruits, such as tomato (*Solanum lycopersicum*, 2–114 ng/g, depending on the variety, the fraction and the method of analysis), cherry (*Prunus avium*, 8–120 ng/g) and grapes (*Vitis vinifera*, 5–96 ng/g) [24].

Modulating blood melatonin levels in mammals through the intake of these products has become a strategy of great interest [26]. In fact, several studies have shown that the consumption of certain melatonin-rich vegetables, seeds and plant products increases the levels of this hormone in blood [27,28]. Given that melatonin readily crosses physiological barriers such as the blood–testis barrier [29], we hypothesized that a phytomelatonin-rich diet could have a beneficial effect on ram semen. In the present study, by-products from food industry derivatives were included in the diet in order to achieve circular economy objectives. Specifically, our aims were to evaluate the effect of the phytomelatonin-rich diet on melatonin concentration and the activity of the antioxidant enzymes in seminal plasma, and on sperm quality of rams.

## 2. Materials and Methods

Unless otherwise stated, all reagents were purchased from Sigma-Aldrich (St. Louis, MO, USA).

### 2.1. Determination of Melatonin Content in Vegetables and Residues after In Vitro Digestion

The melatonin content in several agri-food by-products was analyzed to choose which could potentially be used as a supplement in the animals’ diet. The analyzed products were by-products of the juice (pomegranate pomace and peels), the canning (tomato pomace), the brewing (brewer’s spent grain, malt sprouts and spent yeast) and the wine (grape pulp and grape seeds) industries. Another product with reported high melatonin content, sunflower meal [25,30], was included as a control.

Melatonin was determined following the protocol published by Rebollo-Hernanz et al., 2019 [31]. Briefly, by-products were homogenized until turning them into flour. Melatonin was extracted by treating the flours with MeOH under continuous stirring at 4 °C (16 h in darkness). After centrifugation, the supernatants were filtered under vacuum and dried using N_2_. The residues were resuspended in Milli-Q water and melatonin was isolated using solid-phase extraction (SPE, cartridge C-18, Waters) and measured by HPLC-ESI-MS/MS triple quadrupole. Melatonin content was expressed as ng g^−1^ sample. From each sample, triplicate extractions were made and each one was injected twice into the column.

Melatonin content in the feed residues remaining after in vitro studies or ruminal and abomasal digestion was estimated by the same protocol.

### 2.2. Analysis of Chemical Composition of the By-Products

The by-products selected on the basis of their melatonin content were: grape pulp, tomato pomace, pomegranate peels and pomegranate pomace, to which sunflower meal was added as control.

The AOAC (Association of Official Analytical Chemists) methods [32] were used for the analysis of dry matter (DM, method 934.01), organic matter (OM, method 942.05), crude protein (CP, method 976.05) and ether extract (EE, method 2003.05) content. Concentration of neutral detergent fibre (NDFom) was analyzed as described by Mertens [33] in an Ankom 200 Fibre Analyser (Ankom Technology, New York, NY, USA), using α–amylase and sodium sulphite, with the results being expressed exclusive of residual ashes. The acid detergent fibre (ADF, method 973.18) and acid detergent lignin (ADL) were determined as described by AOAC [32] and Robertson and Van Soest [34], respectively.

### 2.3. Analysis of In Vitro Digestibility of the By-Products

The four selected by-products and the sunflower meal were ground (1 mm particle size), analyzed for chemical composition and incubated in vitro in a closed batch system. Rumen fluid was obtained from four adult, rumen-cannulated ewes. Extraction procedures were approved by the Ethics Committee for Animal Experimentation of the University of Zaragoza (protocol PI48/20). The care and management of animals followed the Spanish Policy for Animal Protection RD 53/2013, which complies with EU Directive 2010/63 on the protection of animals used for experimental and other scientific purposes.

On each incubation run, the rumen contents (approximately 300 mL of each animal) were sampled before feeding, filtered through a cheesecloth, mixed, and immediately transferred to the lab for incubation.

Four in vitro closed batch incubation series were carried out. Incubations were run at 39 °C in a water bath for 24 h under anaerobic conditions following Theodorou et al. [35] with modifications [36]. Five gas bottles were filled with 800 mg of sample sealed in nylon bags and then 80 mL of incubation medium including rumen inoculum (0.2 of total incubation volume) were added. Three bottles without substrate were also included as blanks.

Pressure produced on the bottles was measured with a HD8804 manometer fitted to a TP804 pressure gauge (DELTA OHM, Caselle di Selvazzano, Italy). Readings were corrected for the atmospheric pressure and converted to volume (mL) using a pre-established linear regression (n = 103, *R*^2^ = 0.996) and expressed per unit of incubated organic matter (OM). At the end of the 24 h incubation, 40 mL of the liquid phase and the solid residue were collected from one bottle per treatment and immediately frozen until analysis of the melatonin content. The solid residues of the remaining three bottles were washed with tap water and dried (60 °C, 48 h) to determine ruminal digestibility. Thereafter, dry residues pooled by treatment for each incubation run were digested with HCl-pepsin [37] to estimate abomasal digestibility. The melatonin concentration in the residues after ruminal and abomasal digestion was analyzed as described in Section 2.1.

### 2.4. Animals and Diets

Sixteen 2-year-old Rasa Aragonesa rams were randomly assigned into two groups: eight rams were fed with 500 g of a commercial diet (*Nutrifeed ovejas* 800 MS^®^, Agroveco, Spain) and the other eight were fed with a 500 g phytomelatonin-rich diet per day for five months (from February to July, non-breeding season). The phytomelatonin-rich diet consisted of a mixture of various sources of phytomelatonin (20%), such as pomegranate and tomato pomaces and grape pulp (all of them derivatives of the agri-food industry), which was added to the commercial diet (80%). Diets were formulated to contain the same amount of protein, fat and fibre (Section 2.2). All animals were fed straw ad libitum.

All rams were housed at the Experimental Farm of the University of Zaragoza (Zaragoza, Spain) and all experimental procedures were accomplished as described for the Project License PI39/17 approved by the Ethics Committee for Animal Experiments, University of Zaragoza (Spain), in accordance with the Directive 2010763/UE of the European Parliament on the of animals used for scientific purposes.

### 2.5. Semen Collection and Seminal Plasma Extraction

Ejaculates from each ram were obtained by artificial vagina prior to the beginning of the experiment (month 0, February) and every fifteen days for 5 months (month 5, July), so two complete semen analyses from all animals of each group were performed every month. After semen collection, sperm motility, morphology, membrane integrity, intracellular levels of reactive oxygen species (ROS) and phosphatidylserine (PS) inversion were assessed.

Seminal plasma was obtained by centrifugation at 14,000× *g* for 10 min at 4 °C. The supernatant was collected and centrifuged again under the same conditions, and the recovered seminal plasma was stored at −20 °C until the analysis of the melatonin concentration and the activity of antioxidant enzymes (glutathione reductase, glutathione peroxidase and catalase).

### 2.6. Sperm Motility Analysis

A computer-assisted sperm analysis system (CASA) was used for analyzing sperm motility (ISAS v. 1.04, Proiser S.L., Valencia, Spain). For this assessment, a dilution of each semen sample was made in a medium with the following composition: 0.25 M sucrose, 100 mM EGTA, 0.5 mM sodium phosphate, 50 mM glucose, 100 mM HEPES and 20 mM KOH. A drop of 8 μL of each diluted sample (3 × 10^7^ cells/mL), was placed between a pre-warmed slide and a coverslip and maintained at 37 °C in a heated slide holder during analysis. Spermatozoa were recorded using a video camera (Basler A312f, Basler Vision Components, Exton, PA, USA) mounted on a microscope (Nikon Eclipse 50i, Nikon Instruments Int, Tokyo, Japan) equipped with a 10× negative-phase contrast lens. The recording was performed at 25 frames/s and 25 consecutive digitalized images were taken for a single field. Five fields of each drop were recorded, and percentages of total motile and progressive motile spermatozoa in all samples were evaluated.

### 2.7. Sperm Morphological Study

The sperm morphology was evaluated by eosin-nigrosine staining [38]. A volume of 10 µL of each dye was added to 20 μL of each sample (4 × 10^7^ cells/mL). After mixing, a drop of the stained sample (20 μL) was placed on a slide and spread with the aid of another. The smears were air-dried and observed by bright field microscopy using a Nikon Eclipse E-400 microscope (Kanagawa, Yokohama, Japan). At least 200 spermatozoa were analyzed at 1000× magnification, and the percentage of normal morphology cells was evaluated, considering abnormal those cells that showed primary (detached head) and secondary (bent tail, coiled tail or proximal or distal droplet) abnormalities [38].

### 2.8. Flow Cytometry Analyses

All flow cytometry measurements were performed using a Beckman Coulter FC 500 flow cytometer (Beckman Coulter Inc., Fullerton, CA, USA) equipped with CXP software, two lasers of excitation (argon-ion laser, 488 nm; and solid-state laser, 633 nm) and five filters of absorbance (FL1-525, FL2-575, FL3-610, FL4-675 and FL5-755; ±5 nm each bandpass filter). A flow rate stabilized at 200–300 cells/s was used, and a minimum of 20,000 events were recorded in all experiments. The sperm population was gated for further analysis on the basis of its specific forward (FS) and side scatter (SS) properties and other non-sperm events were excluded.

#### 2.8.1. Evaluation of Sperm Membrane Integrity

To determine cell membrane integrity (viability), a modification of the procedure described by Harrison and Vickers [39] was followed. Samples (500 μL; 5 × 10^6^ cells/mL) were fixed with 3 μL formaldehyde (0.5% (*v*/*v*) in water) and stained with 3 μL of 10 μM carboxyfluorescein diacetate (CFDA) and 3 μL of 7.3 μM propidium iodide (PI). After incubation (15 min, 37 °C in darkness), samples were analyzed by flow cytometry. The monitored parameters were FS log, SS log, FL1 log (CFDA) and FL4 log (PI).

#### 2.8.2. Intracellular Content of ROS

ROS levels were assessed by using 2′,7′-dichlorohydrofluorecein-diacetate (H_2_DCFDA), which is freely permeable across cell membranes [40] and converted into non-permeable and non-fluorescent 2′,7′-dichlorodihydrofluorescein (H_2_DCF) by intracellular esterases. The H_2_DCF is oxidized by H_2_O_2_ to dichlorofluorescein (DCF), which emits fluorescence at 530 nm in response to 488 nm excitation [41]. This probe was combined with PI to exclude the nonviable population from the analysis [42]. For the assessment, sperm samples (final concentration 5 × 10^6^ cells/mL) were stained with 5 μL of 10 μM H_2_DCFDA, and 3 μL of 1.5 mM PI. After 15 min of incubation (37 °C in darkness), samples were fixed with 5 μL formaldehyde (0.5% (*v*/*v*) in water) and analyzed. The monitored parameters were FS log, SS log, FL1 log (H_2_DCFDA) and FL4 log (PI).

#### 2.8.3. Detection of Membrane Phosphatidylserine Translocation

Annexin V is a protein with a high affinity for phosphatidylserine (PS); hence, it can be used as a sensitive probe to detect PS exposure upon the cell membrane [43]. For the analysis, aliquots of 50 μL of sperm samples were diluted (final concentration 4 × 10^6^ cells/mL) with 250 μL of 1X binding buffer (provided in the commercial kit; Binding Buffer Apoptosis Detection Kit, Life Technologies, Carlsbad, CA, USA) and stained with 3 μL of 1.5 mM PI and 2 μL FITC-Annexin V (conjugate also provided in the kit). After 15 min of incubation (37 °C in darkness), samples were assessed by flow cytometry. The monitored parameters were FS log, SS log, FL1 log (Annexin-V) and FL4 log (PI).

### 2.9. Melatonin Evaluation in Seminal Plasma

Melatonin concentration in ram seminal plasma was measured using a commercial competitive immunoassay (Direct saliva melatonin ELISA kit, Bühlmann Laboratories AG, Schönenbuch, Switzerland) as previously described [44]. Absorbance was measured at 450 nm on a microtiter plate reader (SPECTROstar Nano, BMG Labtech, Ortenberg, Germany).

### 2.10. Antioxidant Enzyme Activity Assays in Seminal Plasma

#### 2.10.1. Glutathione Reductase (GRD, EC.1.6.4.2)

All measurements were performed as previously described [16] and all samples were loaded in duplicate. The reaction mixture contained 501.38 mM sodium phosphate buffer at pH 7.2; 0.5 mM EDTA; 85 μM NADPH^+^ + H^+^ and 0.8 mM GSSG. 5 μL of seminal plasma were added to complete a final volume of 200 μL. The enzymatic activity was evaluated for 3 min at 340 nm with a microtiter plate reader (SPECTROstar Nano, BMG Labtech, Ortenberg, Germany).

#### 2.10.2. Glutathione Peroxidase (GPx, EC.1.11.1.9)

Measurements were performed as previously described [16,45] based on a modification of the procedure described by Plagia and Valentine [46]. The reaction mixture contained 501.38 mM sodium phosphate buffer at pH 7.2; 0.5 mM EDTA; 85 μM NADPH^+^ + H^+^, 54 mUI GRD, 2 mM GSH and 1.2 mM t-BuO_2_H. A total of 6 μL of seminal plasma was added to complete a final volume of 200 μL. The absorbance change at 340 nm was monitored for 3 min with the microtiter plate reader (SPECTROstar Nano, BMG Labtech, Ortenberg, Germany).

#### 2.10.3. Catalase (CAT, EC. 1.11.1.6)

For catalase evaluation, the reaction mixture contained 62.5 mM sodium phosphate buffer at a pH 7, 200 mM H_2_O_2_ and 4 μL of seminal plasma to complete a final volume of 200 μL. The absorbance change at 240 nm was monitored for 120 s with the microtiter plate reader (SPECTROstar Nano, BMG Labtech, Ortenberg, Germany). For these measurements, a quartz microplate was used.

### 2.11. Statistical Analyses

Results of in vitro digestion were analyzed statistically by ANOVA with the Statistix 10 package (Analytical Software. Statistix 10 for Windows; Analytical Software: Tallahassee, FL, USA, 2010), considering the incubation run as a block. Treatment differences among means with *p* < 0.05 were accepted as representing statistically significant differences. When significant, differences were contrasted by the Tukey *t*-test.

The chemical compositions of the diets (control vs. phytomelatonin-rich) were compared by the Mann–Whitney test.

Data are shown as mean ± S.E.M. (standard error of the mean). The effects of the diet and time on sperm motility, morphology, viability, intracellular levels of ROS, phosphatidylserine translocation, melatonin concentration and the activities of antioxidant enzymes in seminal plasma were analyzed using the mixed-model ANOVA and Fisher’s LSD as a post hoc test. Statistical analyses were performed with SPSS Statistics v.26 (IBM Analytic, Armonk, NY, USA) and GraphPad Prism v.8 (La Jolla, CA, USA).

## 3. Results

### 3.1. Melatonin Content in Vegetables

According to the HPLC-ESI-MS/MS analysis, the by-products containing the highest levels of phytomelatonin were pomegranate pomace and peels, tomato pomace and grape pulp (Table 1). We therefore selected these by-products for subsequent analysis, together with the sunflower meal.

### 3.2. Melatonin Content after In Vitro Digestion and Diet Composition

The chemical composition of the selected by-products is summarized in Appendix A. After incubation of these by-products in ruminal liquid and the treatment of residues with HCl-pepsin, the melatonin content in the obtained fractions was measured, and the results are shown in Table 2.

The in vitro ruminal and abomasal substrate digestibility and rumen in vitro fermentation pattern of these four by-products and also the sunflower meal are summarized in Appendix A and Appendix A, respectively.

Based on these results, for the in vivo experiment, we selected pomegranate and tomato pomaces and grape pulp mixed in equal proportions. We opted for the pomegranate pomace instead of the pomegranate peels since the rumen fermentation was better (Appendix A), as well as the acceptability by the animals. The mixture accounted for 20% of the daily ration ingested by the treated animals. The chemical analysis of the phytomelatonin-rich and commercial (control) diets (Appendix A) revealed that the differences in CP, EE and fibre were not significant (*p* < 0.05).

### 3.3. Effects of Phytomelatonin-Rich Diets on Seminal Plasma

#### 3.3.1. Effect on Melatonin Levels in Seminal Plasma

Melatonin levels in seminal plasma from rams fed with the phytomelatonin-rich diet showed an increase from month 2, which was statistically significant (*p* < 0.05) in month 3 when compared with rams from the control group. Afterwards, melatonin levels tended to decrease (month 4) and then rise (month 5) both in rams fed with phytomelatonin and rams fed with the commercial diet (Figure 1).

#### 3.3.2. Effect on Antioxidant Enzymes Activity in Seminal Plasma

There were no statistical differences in the activity of catalase, glutathione reductase and glutathione peroxidase between the two experimental groups throughout the duration of the study (Figure 2a, Figure 2b and Figure 2c, respectively).

### 3.4. Effects of Phytomelatonin-Rich Diets on Sperm Quality

#### 3.4.1. Effect on Sperm Motility

For total motility, there was a significant effect of the diet (*p* < 0.01). Percentages of total motile sperm cells in samples from animals fed with the phytomelatonin diet tend to be higher than in the control group samples (Figure 3), and the post hoc test showed that this increment was statistically significant (*p* < 0.01) just in the first month after diet administration. However, there was no significant effect of diet on progressive motility.

#### 3.4.2. Effect on Sperm Morphology

Significant effects of the administered diet were observed on the percentages of morphologically normal spermatozoa (*p* < 0.0001). After month 2, rams fed with the phytomelatonin-rich diet showed higher percentages of morphologically normal cells than the control group until the end of the experiment (Figure 4), with the maximum differences reached at months 2 and 3 (*p* < 0.001).

#### 3.4.3. Effect on Sperm Viability, ROS Levels and PS Translocation

Regarding the percentages of viable, viable with low levels of ROS and viable with no PS translocation spermatozoa, the phytomelatonin-rich diet had a significant effect on sperm viability (*p* < 0.001), and on the content of ROS (*p* = 0.01).

From the second month of the study, sperm viability was significantly higher in rams fed with the phytomelatonin-rich diet than in the control group (*p* < 0.01). These differences were also found until the end of the study, especially in the third and the fifth months (*p* < 0.001, Figure 5a). A similar pattern was observed when analyzing the production of ROS, since rams fed with the phytomelatonin-rich diet had significantly higher percentages of viable sperm with low ROS levels than the control group from month 2, this being more marked at month 3 (*p* < 0.001, Figure 5b). Nevertheless, this effect was not observed when studying the phosphatidylserine inversion, as there were no statistical differences in the percentages of viable spermatozoa without PS translocation between the two groups during the experiment (Figure 5c).

## 4. Discussion

Results of the present experiment have revealed that rams fed with a phytomelatonin-rich diet experience an increase in the content of melatonin in their seminal plasma, improving sperm viability and morphology, and protecting sperm cells against oxidative damage. In previous studies, the beneficial effects of melatonin subcutaneous implants on testis size, ejaculate volume, sperm concentration, motility and morphology, among others, have been reported [15,17,18]. Likewise, melatonin implants increased melatonin levels in seminal plasma during the non-reproductive season [16,19]. However, as consumers are increasingly aware of animal welfare and organic production, it is of interest to adapt animal production to new consumer demands. Replacing the synthetic melatonin used in subcutaneous implants with other natural sources of melatonin, such as phytomelatonin (melatonin from plants), could be a good option, provided that it does not adversely affect the nutritive feed value. In addition, obtaining these sources from by-products of the agri-food industry is consistent with the objectives of the circular economy.

In accordance with the existing bibliography [24,25,26,47] and having regard to the availability of by-products from the food industries in our region, we selected by-products from the wine, beer, juice and canning industries. Analysis of these products by HPLC-ESI-MS/MS revealed that pomegranate pomace and peels, grape pulp and tomato pomace had the highest melatonin content. After evaluating the chemical composition, the in vitro digestibility and the melatonin remaining after in vitro ruminal and abomasal digestion, we selected pomegranate pomace, grape pulp and tomato pomace as the ingredients for a phytomelatonin-rich diet. Pomegranate peels showed higher melatonin content after in vitro digestion than pomegranate pomace but a worse rumen fermentation pattern, digestibility, and acceptability by the animals. Sunflower meal could also have been a good alternative, but in this study it was included only as a control since the work focused only on by-products.

Feeding the rams with a phytomelatonin-rich diet containing a 20% proportion of a mix of by-products (tomato pomace, pomegranate pomace and grape pulp) resulted in an increase in seminal plasma melatonin levels compared with the control group in the third month of the study. However, the use of melatonin subcutaneous implants led to an earlier, higher increase in seminal plasma melatonin levels than that resulting from administering the phytomelatonin-rich diet [16,19]. This could be explained because the quantity of melatonin that can be administered through subcutaneous implants is much higher than the quantity found in the agri-food by-products used in the diet (18 mg in melatonin subcutaneous implants vs. 23.76 ± 1.37, 45.94 ± 4.19 and 35.81 ± 0.40 ng/g for tomato pomace, grape pulp and pomegranate pomace, respectively).

Regarding the possible effects of the diet on sperm quality, parameters including sperm motility, morphology, membrane integrity, intracellular ROS levels and PS translocation were evaluated. We observed that total motility tends to be higher in semen collected from animals fed with the phytomelatonin-rich diet than in the control group samples, although this increase was only significant in the first month after administering the diet. This finding is in agreement with results previously described for Black Racka rams treated with subcutaneous melatonin implants during the non-breeding season, since treated animals from this breed showed better total sperm motility rates than the control group [17]. However, feeding a phytomelatonin-rich diet had no significant effect on progressive motility, contrary to what was described when melatonin implants were used in rams of the same [48] or other breeds [15,17].

The phytomelatonin-rich diet led to higher percentages of morphologically normal and viable spermatozoa than in the control group from the second month to the end of the experiment. Beneficial effects on morphology were also reported in some studies using melatonin implants [15] but not others [17]. In the case of viability, no changes were reported after the use of melatonin implants in rams [15], but beneficial effects were described in other species, including bull [23], buck [22] and buffalo [21]. In ovine, improvement in sperm viability was observed when melatonin was added directly to spermatozoa in vitro [49,50]. However, in this work, we describe, for the first time, how melatonin supplemented through diet can enhance the percentage of morphologically normal and membrane-intact sperm cells.

A similar pattern occurred with the percentage of live sperm cells with low content of reactive oxygen species (ROS). It has been demonstrated that in vitro incubation of sperm samples with melatonin can reduce oxidative stress damage caused by ROS in human and boar spermatozoa [50,51]. Recently, our research group has documented that melatonin, when added to sperm samples in capacitating conditions, can reduce ROS levels and partially prevent sperm capacitation [9]. However, in the present study, no significant differences in the percentages of viable spermatozoa without inversion of phosphatidylserine were observed between the samples from the two groups. Although in vitro melatonin has an antiapoptotic effect on sperm from different species [51,52], including ovine [49], its oral administration through the assayed diet does not seem to influence this parameter in rams.

According to the data presented, we can infer that a diet rich in phytomelatonin can protect sperm cells against oxidative damage by decreasing intracellular ROS levels. This antioxidant effect could be exerted directly or indirectly by influencing the activities of antioxidant enzymes [53]. In the present study, the analysis of the activities of catalase (CAT), glutathione reductase (GRD) and glutathione peroxidase (GPX) in seminal plasma revealed no significant differences between the two experimental groups. Although melatonin implants modulate the activity of GPX and GRD, but not CAT, in rams [16] and bucks [20], the tested phytomelatonin-rich diet seems not to modify the activities of these antioxidant enzymes, probably because the amount of exogenous melatonin provided with the diet is much smaller than that produced by implants [16]. As no changes in the activities of the aforementioned antioxidant enzymes were detected, we hypothesized that phytomelatonin could modulate the levels of reactive oxygen species directly in the spermatozoa, either by crossing the sperm plasma membrane or by binding to its specific membrane receptors. Nevertheless, the molecular mechanisms by which the phytomelatonin-rich diet exerts this antioxidant action on spermatozoa remain unknown. Furthermore, we cannot overlook the fact that other antioxidants present in the by-products used in the diet may contribute synergistically with phytomelatonin to produce this effect. For instance, polyphenols and anthocyanins present in pomegranate have been shown to have powerful antioxidant activity [54], as well as lycopene, ascorbic acid, vitamin E and flavonoids present in tomato [55], or catechins, flavonols, benzoic acid and cinnamic acid present in grape pulp [56].

## 5. Conclusions

This study reveals, for the first time, that a phytomelatonin-rich diet increases melatonin levels in seminal plasma, improves sperm viability and morphology, and protects sperm cells against oxidative damage by decreasing intracellular ROS levels; all these effects occurring, at the latest, three months after the beginning of the feeding.

## Figures and Tables

**Figure 1 animals-13-00905-f001:**
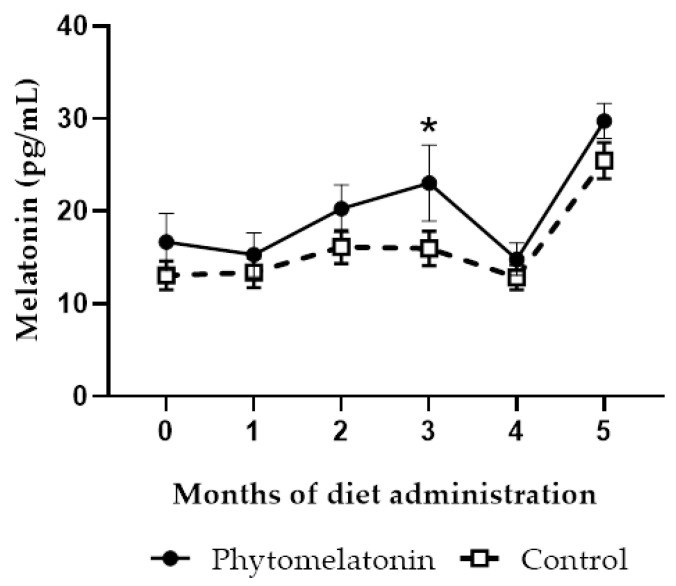
Monthly variations of melatonin levels in seminal plasma of animals fed with a phytomelatonin-rich (●) or commercial (control, □) diet from month 0 (before diet administration) to month 5. Values are shown as mean ± S.E.M., n = 16. * *p* < 0.05 indicates significant differences between both experimental groups.

**Figure 2 animals-13-00905-f002:**
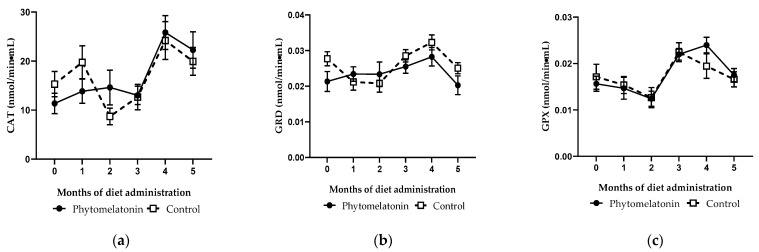
Monthly variations of catalase (CAT), (**a**), glutathione reductase (GRD), (**b**) and glutathione peroxidase (GPX), (**c**) activities in seminal plasma of animals fed with a phytomelatonin-rich (●) or commercial (control, □) diet from month 0 (before diet administration) to month 5. Values are shown as mean ± S.E.M., n = 16.

**Figure 3 animals-13-00905-f003:**
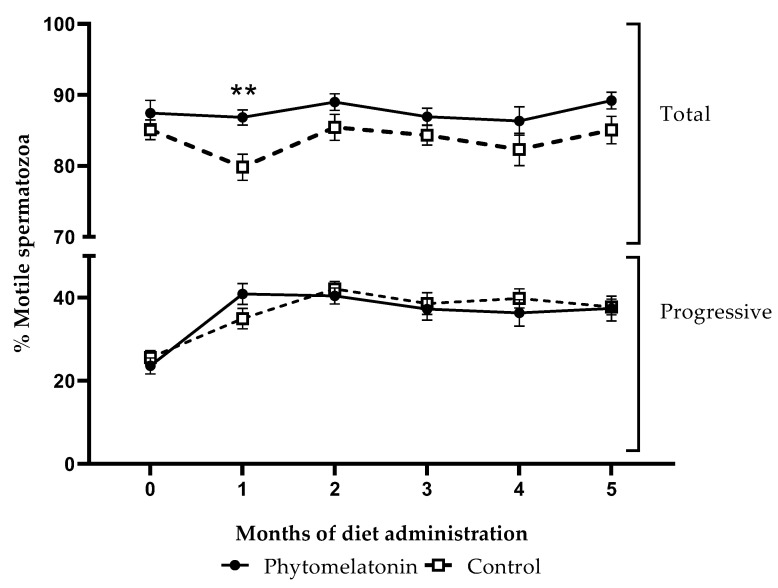
Percentage of total motile (top) and progressive (bottom) spermatozoa in semen samples from animals fed with a phytomelatonin-rich (●) or commercial (control, □) diet from month 0 (before diet administration) to month 5. Values are shown as mean ± S.E.M., n = 16. ** *p* < 0.01 indicates significant differences between the two experimental groups.

**Figure 4 animals-13-00905-f004:**
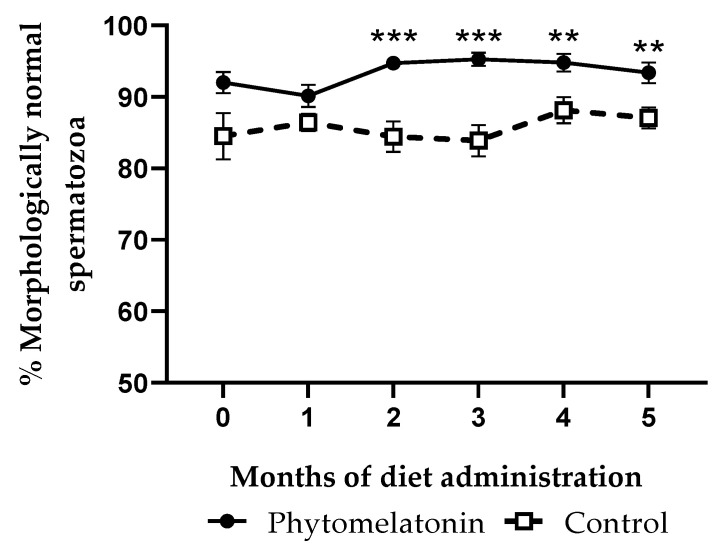
Percentage of sperm cells with normal morphology in samples from animals fed with a phytomelatonin-rich (●) or commercial (control, □) diet from month 0 (before diet administration) to month 5. Values are shown as mean ± S.E.M., n = 16. ** *p* < 0.01 and *** *p* < 0.01 indicate significant differences between both experimental groups.

**Figure 5 animals-13-00905-f005:**
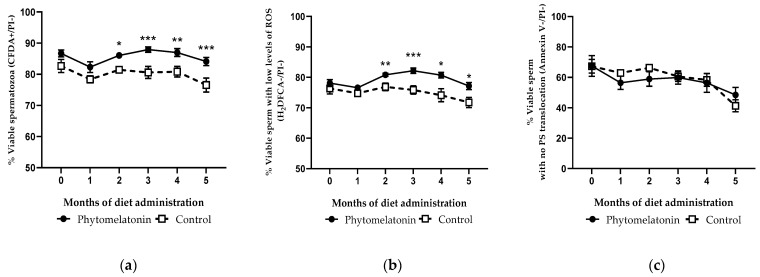
Percentage of viable (**a**), viable with low ROS levels (**b**) and viable without phosphatidylserine (PS) translocation (**c**) spermatozoa in samples from animals fed with a phytomelatonin-rich (●) or commercial (control, □) diet from month 0 (before diet administration) to month 5. Values are shown as mean ± S.E.M., n = 16. * *p* < 0.05, ** *p* < 0.01 and *** *p* < 0.01 indicate significant differences between the two experimental groups.

**Table 1 animals-13-00905-t001:** Melatonin content in agri-food by-products analyzed by HPLC-ESI-MS.

Sample	Melatonin (ng/g)
Pomegranate pomace	35.81 ± 0.40
Pomegranate peels	127.03 ± 8.73
Tomato pomace	23.76 ± 1.37
Brewer’s spent grains	3.01 ± 0.15
Malt sprouts	3.33 ± 0.04
Spent yeast	5.03 ± 0.11
Grape pulp	45.94 ± 4.19
Grape seeds	9.26 ± 0.52
Sunflower meal	41.55 ± 5.16

**Table 2 animals-13-00905-t002:** Melatonin content in by-products after in vitro digestion.

Sample	Melatonin Content (ng/g) *		
	After Ruminal Incubation		After HCl-Pepsin Treatment
	Liquid Fraction	Solid Fraction	
Pomegranate pomace	0.53 ± 0.04	n.d.	0.93 ± 0.58
Pomegranate peels	1.56 ± 0.47	0.62 ± 0.14	n.d.
Tomato pomace	2.69 ± 0.69	0.32 ±0.02	0.54 ± 0.11
Grape pulp	1.58 ± 0.19	0.34 ± 0.12	n.d.
Sunflower meal	1.22 ± 0.15	0.58 ± 0.06	0.32 ± 0.08

* Melatonin content expressed on dry matter of residue (mean ± SEM) (n = 4 corresponding to four incubation bottles). n.d.: not detected.

## Data Availability

The datasets generated for this study can be found in the figshare repository doi: 10.6084/m9.figshare.22193245.

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
