# Peer review of "Improvement of the Seminal Characteristics in Rams Using Agri-Food By-Products Rich in Phytomelatonin"

_animals, 2023, doi:10.3390/ani13050905_

Round 1
Reviewer 1 Report
2.4. Animals and diets. How many animals were there at the beginning of the study? 16 rams from the start or was there a bigger group? I don't understand the randomization method.
Were the animals healthy before being included in the study group? Were they examined in terms of the reproductive system, e.g. ultrasound of the testicles? Have these males mated before? Vaccinated, dewormed etc...
2.5. Semen collection and seminal plasma extraction. Was an electroejaculator used?
Figure 1. How to explain the increase up to 3 months and then the decrease? Has the animal-administered extract been tested for mycotoxins? Could a given batch differ from each other?
Regards
Author Response
Thank you very much for your review. We have tried to answer all your questions.
2.4. Animals and diets. How many animals were there at the beginning of the study? 16 rams from the start or was there a bigger group? I don't understand the randomization method.
For this study, we acquired a total of 24 young rams, and after training for several months, we were able to obtain ejaculates regularly with the aid of an artificial vagina from 18 of them. The seminal quality of each of them (motility and viability) was analyzed during training. Before the start of the experiment, a preliminary palatability test of the diet was carried out. Finally, two groups of 8 animals were made, taking into account that the semen quality of each group was similar and that the animals in the experimental group (phytomelatonin) consumed the diet properly.
Were the animals healthy before being included in the study group? Were they examined in terms of the reproductive system, e.g. ultrasound of the testicles? Have these males mated before? Vaccinated, dewormed etc...
Yes, the animals were healthy before and during the study. Rams were housed in the facilities of the Veterinary School (University of Zaragoza) under the supervision of the Servicio General de Apoyo a la Experimentación Animal (SAI), which was in charge of their feeding and care, including vaccination and deworming. Also, ultrasonography evaluation of the testis was carried out using a portable ultrasound scanner (ExaGo, IMV imaging Angoulême, France) connected to a 7.5 MHz linear probe. The rams were restrained, and no sedatives were used. The probe was positioned transversely to the major axis of the testicle. Three videos of 124 frames each were recorded in the upper, medium and lower parts of each testicle for echotexture analyses. The echotexture analysis was performed with ECOTEXT® software (Humeco, Huesca, Spain). These evaluations were performed before and monthly throughout the experiment. Results were not included in this manuscript.
These males arrived at our facilities when they were 5-6 months old and had not mated before. During the training period, they were all together in the same yard, so we cannot rule out mating between them. During the study, they were housed individually to avoid mating with each other, facilitate handling and control that they consumed the diet adequately.
2.5. Semen collection and seminal plasma extraction. Was an electroejaculator used?
As described in section 2.5, the semen was obtained using an artificial vagina. A sheep in heat was used as a stimulus. The animals were trained for 2-3 months to get used to semen collection in an artificial vagina. No electroejaculator was used at any time.
Figure 1. How to explain the increase up to 3 months and then the decrease? Has the animal-administered extract been tested for mycotoxins? Could a given batch differ from each other?
The melatonin in the seminal plasma, at least of rams located in mid-latitude zones, has both a pineal and testicular origin (Carvajal-Serna et al. (2020). "Vasectomy and Photoperiodic Regimen Modify the Protein Profile, Hormonal Content and Antioxidant Enzymes Activity of Ram Seminal Plasma." International Journal of Molecular Sciences 21(21): 8063). While melatonin synthesis in the testicles would be independent of the duration of light exposure, that from the pineal gland does depend on the photoperiod. Month 4 corresponded to June, which is the month with the shortest nights, and therefore the secretion of nocturnal pineal melatonin is lower. This would result in a lower amount of melatonin in the seminal plasma, which is evident in the two experimental groups (Control and Phytomelatonin).
The extract was not tested for mycotoxins but no animal showed signs of intoxication throughout the study.
The by-products and the mixture were from the same batch throughout the entire study.
Reviewer 2 Report
It was a very nice and interesting article to read; please consider the following edits. Thank you
Line 44: I suggest starting the introduction with the importance of sheep in your study area/region or country. Please restructure the first two paragraphs of the introduction. A paragraph cannot be of two sentences (lines 45- 48).
Line 49-52: "This horses,” Please state the name of the hormone as it is the new paragraph. Again, this paragraph is one sentence. Please restructure your first and second paragraphs of the introduction as a paragraph should stay around one topic, and it should not be one or two sentences.
Line 53-54: Please cite your statement. And please tell the reader more about those findings and what were their conclusion/significant results.
Line 53-56: Again, a two-sentence paragraph and not give adequate information about the findings that you are pointing to in previous studies.
Line 57-58: Please cite.
Line 67-68: Please cite.
Line 79-85: Please state the values of melatonin concentration for each plant separately.
Line 92-98, 90-91, 55-56: You wrote your aim several times in the introduction, which is not necessary. The aims presented in the summary, abstract, and at the end of the introduction are enough. Please edit accordingly.
Line 180-182: A sample of 8 for the control and treatment group. How you calculated the sample size? And is it enough to generalize the results? Please support your answer with previously published literature.
Line 432-435: Did you check the palatability of your mix of by-products (tomato pomace, pomegranate pomace, and grape pulp)?
Para (line 463-470) and para (471-475) should be concise, and I suggest making one paragraph of these two paragraphs as they both discuss one topic.
Line 499: I suggest not starting the conclusion with vague words like "to sum up". This is not a summary but a conclusion.
Author Response
Thank you very much for your comments. We have tried to consider all your suggestions.
Line 44: I suggest starting the introduction with the importance of sheep in your study area/region or country. Please restructure the first two paragraphs of the introduction. A paragraph cannot be of two sentences (lines 45- 48).
Thank you for your suggestions. We have now included the sentence: “Livestock farming systems in the Mediterranean regions of the southern European Union countries, such as sheep husbandry, are important [1] given that they are linked with the use of semi-natural and natural areas, and involve well-adapted autochthonous breeds [2], which is the case for the Rasa Aragonesa breed [3]” with the corresponding bibliographic references. Also, the first paragraphs of the introduction have been rewritten and restructured. (current lines 45-55 in the revised manuscript)
Line 49-52: "This horses,” Please state the name of the hormone as it is the new paragraph. Again, this paragraph is one sentence. Please restructure your first and second paragraphs of the introduction as a paragraph should stay around one topic, and it should not be one or two sentences.
This part of the introduction has been restructured and included in the first paragraph (current lines 51-55)
Line 53-54: Please cite your statement. And please tell the reader more about those findings and what were their conclusion/significant results. Line 53-56: Again, a two-sentence paragraph and not give adequate information about the findings that you are pointing to in previous studies.
This paragraph (lines 53-56 in the original manuscript) has now been deleted as it referred to the findings mentioned in the previous sentence, where the bibliography was cited. Also, the mentioned aim was redundant, as the reviewer points out below.
Line 57-58: Please cite.
This statement (lines 57-58 in the original manuscript) was an introduction to the sentences that followed, where the bibliographical references were included, but following the reviewer's advice, we have included the citations here (current lines 56-57).
Line 67-68: Please cite.
This statement (lines 67-68 in the original manuscript) was an introduction to the sentences that followed, where the bibliographical references were included, but following the reviewer's advice, we have included the citations here (current lines 66-68).
Line 79-85: Please state the values of melatonin concentration for each plant separately.
The values of melatonin concentration for each plant have been added (current lines 78-85).
Line 92-98, 90-91, 55-56: You wrote your aim several times in the introduction, which is not necessary. The aims presented in the summary, abstract, and at the end of the introduction are enough. Please edit accordingly.
We have removed the sentences written in Iines 55-56 and 92-93, and this part of the introduction has been edited accordingly (current lines 89-95).
Line 180-182: A sample of 8 for the control and treatment group. How you calculated the sample size? And is it enough to generalize the results? Please support your answer with previously published literature.
For this study, we acquired a total of 24 young rams, and after training for several months, we were able to obtain ejaculates regularly with the aid of an artificial vagina from 18 of them. The seminal quality of each of them (sperm motility and viability) was analyzed during training. Before the start of the experiment, a preliminary palatability test of the diet was carried out. Finally, two groups of 8 animals were made, taking into account that the semen quality of each group was similar and that the animals in the experimental group (phytomelatonin) consumed the diet properly.
Regarding the sample size, we did not know the effect size or the standard deviation to use a power analysis. Thus, we used the “Resource equation method” (Mead R. 1988. The design of experiments. Cambridge, New York: Cambridge University Press. 620 p). With this method, any sample size that keeps E between 10 and 20 should be considered adequate. The following formula can measure E:
E = Total number of animals − Total number of groups= 16 – 2=14, which is between 10 and 20 and indicates that we had enough animals in our experimental groups.
The following papers include more information on the Resource equation: Festing MF, Altman DG. Guidelines for the design and statistical analysis of experiments using laboratory animals. ILAR J. 2002;43:244–58; How to calculate sample size in animal studies? Charan J, Kantharia ND J Pharmacol Pharmacother. 2013 Oct; 4(4):303-6.
Of course, a higher number of animals would give greater robustness to the results, but we must try to minimize the number of experimental animals used, following the recommendations contained in the current Directive 2010/63/UE on the protection of animals used for scientific purpose, and the 3Rs guidelines.
Line 432-435: Did you check the palatability of your mix of by-products (tomato pomace, pomegranate pomace, and grape pulp)?
As mentioned above, a preliminary palatability test of the phytomelatonin-rich diet was carried out and finally two groups of 8 animals were made, taking into account that the semen quality of each group was similar and that the animals in the experimental group (phytomelatonin) consumed the diet properly.
Para (line 463-470) and para (471-475) should be concise, and I suggest making one paragraph of these two paragraphs as they both discuss one topic.
The paragraph (lines 463-470 in the original manuscript) has been resumed, making one paragraph with the following one (current lines 427-437 in the revised manuscript).
Line 499: I suggest not starting the conclusion with vague words like "to sum up". This is not a summary but a conclusion.
The expression “to sum up” has been removed.
Round 2
Reviewer 2 Report
.